# Understanding the Relationship between Urban Biophysical Composition and Land Surface Temperature in a Hot Desert Megacity (Saudi Arabia)

**DOI:** 10.3390/ijerph20065025

**Published:** 2023-03-12

**Authors:** Abdullah Addas

**Affiliations:** 1Department of Civil Engineering, College of Engineering, Prince SattamBin Abdulaziz University, Alkharj 11942, Saudi Arabia; a.addas@psau.edu.sa or aaddas@kau.edu.sa; 2Landscape Architecture Department, Faculty of Architecture and Planning, King Abdulaziz University, P.O. Box 80210, Jeddah 21589, Saudi Arabia

**Keywords:** urban heat island, megacity, urban biophysical composition, principal component analysis, eco-environmental quality

## Abstract

The deteriorations of the thermal environment due to extreme land surface temperature (LST) has become one of the most serious environmental problems in urban areas. The spatial distribution of urban biophysical composition (UBC) has a significant impact on the LST. Therefore, it is essential to understand the relationship between LST and biophysical physical composition (BPC) to mitigate the effects of UHIs. In this study, an attempt was made to understand the relationship between LST and BPC in a hot desert coastal megacity (Jeddah megacity) in Saudi Arabia. Principal component analysis (PCA) was used to understand the factors affecting LST based on remote sensing indices. Correlation and regression analyses were carried out to understand the relationship between LST and BPC and the impact of BPC on LST. The results showed that, in Jeddah city from 2000 to 2021, there was a substantial increase in the built-up area, which increased from 3085 to 5557.98 hectares. Impervious surfaces had a significant impact on the LST, and green infrastructure (GI) was negatively correlated with LST. Based on the PCA results, we found that the GI was a significant factor affecting the LST in Jeddah megacity. The findings of this study, though not contributing to further understanding of the impact of BPC on LST, will provide planners and policy makers with a foundation for developing very effective strategies to improve the eco-environmental quality of Jeddah megacity.

## 1. Introduction

The world has experienced rapid urban expansion in the last few decades [1,2]. Thus, land use and land cover change (LULC) due to rapid urban expansion has emerged as one of the most significant drivers for the increase in land surface temperature (LST) in urban areas [3,4,5]. LULC has been recognized as one of the main causes of increased LST [6,7]. LULC largely affects the biophysical conditions of the relevant area [8,9,10]. Moreover, the rapid rate of urbanization has led to significant transformations of natural landscapes, such as vegetation cover and waterbodies, into impervious surfaces [11,12]. This phenomenon ultimately affects urban ecosystems, local climates, and energy flow [13]. Thus, understanding the relationship between biophysical indices and LST is crucial for mitigating local climate impacts in cities [14,15]. One of the most serious problems in urban areas area is an increase in temperature due to conversion of the natural landscapes (such as vegetation cover and water cover) into impervious surface cover [16] as well as the conversion of vegetation cover into agricultural and open land. Previous studies have shown that LULC resulting from the conversion of natural to artificial lands has substantial impacts on the LST [17,18]. LULC includes both physical and biological components on the Earth’s surface, and LST is closely linked to the spatial distribution of the physical and biophysical components over the surface [19]. The physical and biological components have different moisture, thermal, and radiative properties, and these properties are strongly linked with the surrounding environment. Thus, LST has a significant connection with the soil moisture, greenness, and wetness of surfaces [20,21]. Continuous urban expansion causes deterioration of waterbodies and vegetation cover, leading to significant alterations in the biophysical environments of urban areas [22,23]. Therefore, it is essential to understand the interrelationships between biophysical components and the urban environment [24]. Thus, in this study, an attempt was made to understand the relationship between the biophysical environment and LST.

In the context of Saudi cities, unprecedented urban expansion with a rapid increase in impervious surfaces could impact the LST and lead to deterioration of the urban thermal environment [25,26,27]. The application of remote sensing indices offers significant opportunities to understand the relationship between biophysical indices and LST [28,29]. In previous literature, a number of remote sensing technologies were used to estimate the impact of the biophysical environment on LST through such indices as the normalized difference vegetation index (NDVI), normalized difference built-up index (NDBI), normalized difference bareness index (NDBaI), normalized difference water index (NDWI), and modified normalized difference water index (MNDWI) [30,31,32]. With this background, a fundamental question emerges: why does this study focus on the LST as one of the most significant parameters to determine impacts on biophysical parameters? LST is considered a significant component in enhancing the quality of urban health [33,34,35]. Increases in impervious surfaces and the deterioration of the natural landscapes result in an increase in the LST and a deterioration of the thermal comfort level in urban environments [36,37,38]. According to Yao et al. [39], LST is considered one of the most significant parameters for monitoring urban ecological quality. The NDVI, NDBI, MNDWI, and NDWI have a close association with LST [19,40]. A continuous increase in LST ultimately leads to the urban heat island (UHI) phenomenon [41,42]. In previous studies, it was recognized that temperatures are higher in urban than rural areas [43,44]. This phenomenon is known as the urban heat island (UHI) effect [43]. The UHI phenomenon is also closely related to the urban biophysical composition (UBC) [44]. Therefore, it is important to understand the relationship between LST and biophysical composition, particularly in a hot desert climate similar to that in Saudi Arabia. 

In this study, Jeddah megacity in Saudi Arabia was analyzed to understand the relationship between LST and biophysical indices. After a quick review of previous research studies, a few notable research gaps were identified. Firstly, most studies in Saudi Arabia have focused on the relationship between LULC and LST, whereas very few sought to understand the relationship between biophysical indices and LST. Secondly, biophysical indices not only help in understanding the urban thermal environment but also play a significant role in understanding the eco-environmental quality in urban areas. In many previous studies, biophysical indices were used for eco-environmental modeling, such as the modeling of ecological quality [45,46], ecological vulnerability [47,48], and environmental quality [49]. Therefore, studying the relationship between biophysical parameters and LST using remote sensing indices will not help us understand the thermal environmental quality but rather the eco-environmental quality of the city. Thirdly, Saudi Arabia is located in a hot desert climate where the temperature during summer reaches above 40 °C. Therefore, it is essential to understand the relationship between LST and biophysical indices for policy implications. However, very few studies focus on this factor. These research gaps highlight the need to conduct a study on the relationship between LST and biophysical parameters to achieve sustainable urban environmental development for human wellbeing.

## 2. Materials and Methods

### 2.1. Study Area

In recent years, Saudi cities have experienced rapid growth in population due to large-scale migration. Cities such as Jeddah, Dammam, and Riyadh have experienced rapid urban expansion in the last 40 years [49]. For this study, Jeddah megacity was selected to understand the relationship between LST and urban biophysical indices Figure 1. Jeddah is one of the most populous megacities in Saudi Arabia and is located in the western part of the country on the western coast of the Red Sea. The total geographical area of this megacity is about 1600 km^2^, with a population density of 5400 person/km^2^. This megacity has a dry and hot desert climate with a maximum temperature above 40 °C during summer and average rainfall of approximately 45 mm.

### 2.2. Data Source

In this study, multi-temporal satellite images were used to create LULC maps and biophysical indices. LULC maps were developed for the years of 2000 and 2021, and both LANDSAT 5 TM and LANDSAT 8 OLI were used. The biophysical indices were developed for the year 2021 using LANDSAT 8 OLI images. Data related to the LULC and biophysical indices were extracted from the United State Geological Survey (USGS) with a spatial resolution of 30 m. Details of the data sources are presented in Table 1.

#### 2.2.1. Image Preprocessing

In this study, the collected data were preprocessed in the arcGIS environment (version 10.3). Bands were extracted from the satellite images to develop the biophysical indices, including NDVI, NDBI, and NDWI. NDBaI, an LST map, was developed from the thermal band (band 10 for LANDSAT OLI).

#### 2.2.2. Extraction of Land Surface Temperature (LST)

LST is one of the most significant parameters for representing the thermal condition in any area [50]. Particularly in urban areas, assessment of the LST is crucial to understanding the spatial patterns of LST and implementing effective strategies for its management [51,52]. There are a few basic steps necessary to extract LST from satellite images. These steps are as follows.

Step 1: Conversion of the digital number to spectral radiance (*L_λ_*)

Every object with a temperature above 0 K emits thermal electromagnetic energy [53]. Radiance from the sensors can be received from the thermal sensors [54]. Thus, the spectral radiance (*L_λ_*) can be calculated using the following equation:(1)$$L_{\lambda }=L_{min\lambda }\left[\frac{L_{max\lambda }-L_{min\lambda }}{Q_{CALmax}-Q_{CALmin}}\times Q_{CAL}\right]$$
where *L* refers to the spectral reflectance derived from the sensor; $L_{max\lambda }$ and $L_{min\lambda }$ refer to the maximum and minimum spectral radiance from the thermal band (6 for TM and 10 for OLI 8); $Q_{CAL}$ refers to the digital number (DN) of the pixel; and $Q_{CALmax}$ and $Q_{CALmin}$ refer to the maximum (255) and minimum (0) DN values, respectively.

Step 2: Converting spectral radiance (*L_λ_*) to brightness temperature (*T_β_*):

In the second step, it is necessary to transform the spectral radiance (*L_λ_*) to reflectance to correct the emissivity. As per the estimations of [55], the vegetated areas were attributed a value of 0.92, and non-vegetated areas were 0.95. The emissivity can be extracted from the following equation:(2)$$T_{\beta }=\frac{K_{2}}{In\left(\frac{K_{1}}{L_{\lambda }}+1\right)}-273.15$$
where $T_{\beta }$  refers to the brightness temperature (*K*); $L_{\lambda }$  refers to the spectral radiance of the sensor (Wm^−2^sr^−1^μm^−1^); and *K*_1_ and *K*_2_ refer to the calibration constant ($K_{1}$ = 60.776 $mW{cm}^{-m}{sr}^{-r}{\mu m}^{-m}$ and $K_{2}$ = 1260.56 for the LANDSAT band). Absolute zero is used to convert the temperature into degrees Celsius.

Step 3: Correction of the emissivity through *NDVI* (*P_v_*)

In step 3, it is necessary to correct the spectral emissivity of the retrieval temperature value. Spectral emissivity is calculated using *NDVI*; the proportion of *NDVI* can then be calculated (*P_v_*). The following equation is used to calculate *P_v_*:(3)$$P_{v}={\left(\frac{NDVI-{NDVI}_{soil}}{{NDVI}_{veg}+{NDVI}_{soil}}\right)}^{2}$$
where ${NDVI}_{soil}$  and ${NDVI}_{veg}$  refer to the soil and vegetation pixel values, respectively. The threshold values of soil and vegetation are 0.70 and 0.20 [56].

Step 4: Emissivity (δ )

Land surface emissivity (LSE) is important to estimate LST and is considered as the proportionality factor of Plank’s law. The LSE is calculated using the following equation:(4)$$\delta_{\lambda }=\delta_{veg.\lambda }P_{v}+\delta_{soil}\left(1-P_{v}\right)+C_{v}$$
where $\delta_{veg}$ and $\delta_{veg}.\lambda$ refer to the vegetation and soil, respectively, and *C* refers to the surface representation.

Step 5: Land surface temperature (*LST*):

Finally, *LST* can be calculated using the following equation:(5)$$LST=\frac{T_{\beta }}{\left[1+\left\{\left(\lambda .\frac{T_{\beta }}{\rho }\right)In.\delta_{\lambda }\right\}\right]}$$
where *LST* refers to the land surface temperature in degrees Celsius; $T_{\beta }$  refers to the brightness temperature of the sensor; *λ* refers to the wavelength of the emitted radiance (λ = 10.895); and $\delta_{\lambda }$  refers to the emissivity. The emissivity is calculated using the following equation:(6)$$\rho =h\frac{C}{\alpha }=1.438\times 10^{-2}mK$$
where α  refers to the Boltzmann constant, which is $1.38\times 10^{-23}\;{JK}^{-1}$; *h* refers to Planck’s constant, which is $6.626\times 10^{-34}\;{JK}^{-1}$; and *C* refers to the velocity of light ($2.998\times 10^{-8}\;{ms}^{-1}$).

#### 2.2.3. Accuracy Assessment

In this study, the accuracy levels of the LULC maps were calculated for the years of 2000, 2010, and 2021. The accuracy of the LULC maps indicates the difference between classified maps and reference data. The overall accuracy (*OA*), user accuracy (*UA*), producer accuracy (*PA*), and kappa statistics derived from the error matrix were used to determine the accuracy level of the classified LULC maps. The following equation is used to derive the accuracy level of the classified LULC maps:$$OA=\frac{Number\;of\;the\;true\;positive\;+\;Number\;of\;the\;true\;negative}{Pixel\;in\;the\;groud\;truth}$$

*UA* measures the commission error representing the probability of the classified pixels over the ground, and *PA* represents the fit of the classification.
$$UA=\frac{{Row\;elements}_{diagonal}}{{Row}_{total}}\;PA=\frac{{Column\;elements}_{diagonal}}{{Column}_{total}}$$

Finally, the kappa coefficient is used to measure the accuracy of the classified LULC images as follows:$$K=\frac{{Row\;elements}_{diagonal}}{{Row}_{total}}.$$

#### 2.2.4. Extraction of the Biophysical Parameters

In this study, the impact of the biophysical indices on the LST was assessed. These indices were NDVI, NDWI, NDBI, NDBaI, and MNDWI. In previous studies, several biophysical parameters were analyzed to understand their impact on the LST, but these studies failed to comprehensively capture the thermal patterns and their impacts [42]. Therefore, it remains essential to analyze the impacts of biophysical parameters on the thermal environment in urban areas [57]. Although an individual spectral parameter can quantify surface characteristics, it is difficult to comprehensively assess the surface characteristics without using a variety of thermal properties related to the biophysical parameters. Thus, in this study, a number of spectral indices were used to characterize the impact of biophysical parameters on the thermal patterns in Jeddah megacity. The details of the biophysical indices are discussed in the following sections:

##### Normalized Difference Vegetation Index (NDVI)

The vegetation status of any area can be represented by the NDVI, which indicates the health of the vegetation [58]. Thus, NDVI is a very useful parameter to quantify the greenness of any region [59]. In previous studies, NDVI was widely used as a significant parameter for analyzing urban growth patterns and microclimatic conditions in urban areas. The value of NDVI ranges from −1 to +1, where a value close to +1 represents healthy vegetation, and a value close to −1 indicates poor vegetation health status. A high value of NDVI indicates vegetation cover, a low positive value indicates built-up and bare lands, and a negative value of NDVI indicates water bodies [60]. The following equation is used to calculate *NDVI*:(7)$$NDVI=\frac{NIR-R}{NIR+R}$$
where *NIR* (near infrared band) indicates band 4 for LANDSAT TM and band 5 for OLI 8, whereas *R* (red band) indicates band 3 for LANDSAT TM and band 4 for OLI 8, respectively.

##### Normalized Difference Water Index (NDWI)

NDWI is the remote sensing parameter used to denote the water level in any area. The value of *NDWI* ranges from −1 to +1, representing low and high water levels, respectively. The value of the *NDWI* is more significant than *NDVI*, mainly due to a lack of sensitivity to atmospheric effects [51]. The following equation is used to calculate *NDWI*:(8)$$NDWI=\frac{G-NIR}{G+NIR}$$
where *G* (green band) represents bands 2 and 3 for LANDSAT 5 and OLI 8, whereas NIR (near infrared band) represents bands 4 and 5 for LANDSAT 5 and OLI 8, respectively.

##### Normalized Difference Built-Up Index (NDBI)

The nature of imperviousness is one of the most significant parameters affecting thermal patterns in urban environments. *NDBI* is a common remote sensing parameter used to denote built-up areas or impervious surface areas in any area. The middle infrared (*MIR*) and near infrared (*NIR*) band values are used to calculate the *NDBI* for impervious surface extractions. The following equation is used to extract *NDBI*:(9)$$NDBI=\frac{MIR-NIR}{MIR+NIR}$$
where *MIR* (middle infrared band) corresponds to bands 5 and 6 for LANDSAT TM and OLI 8, whereas NIR (near infrared bans) corresponds to bands 4 and 5 for LANDSAT TM and OLI 8, respectively. The value of NDBI ranges from −1 to +1, with a value close to ‘0′ indicating vegetation cover; a negative value indicates waterbodies, and a positive value represents impervious surface areas (i.e., built-up areas).

##### Modified Normalized Difference Vegetation Index (MNDWI)

*MNDWI* is one of the most significant remote-sensing-based parameters used to estimate waterbodies without vegetation noise and built-up areas. In previous studies, *MNDWI* was widely used to estimate waterbodies and to understand their impacts on the thermal environment in urban areas [61]. The following equation is used to estimate the *MNDWI*:(10)$$MNDWI=\frac{G-MIR}{G+MIR}$$
where *G* (Green band) represents bands 2 and 3 for LANDSAT 5 and OLI 8, whereas MIR (middle infrared band) represents bands 5 and 6 for LANDSAT TM and OLI 8, respectively.

In this study, correlation analysis was carried out after normalization of the parameters. The following equation was used for the normalization of the parameters used in this study:(11)$$\mathrm{Positive}=\frac{A_{-}A_{min}}{A_{max}-A_{min}}$$
(12)$$\mathrm{Negative}=\frac{{A_{max}}_{-}A}{A_{max}-A_{min}}$$
where A indicates the actual value of the parameter, and A_max_ and A_min_ refer to the maximum and minimum value of the parameter.

### 2.3. Statistical Analysis

In this study, descriptive (mean and standard deviation) and inferential (correlation and regression) statistical analyses were carried out to understand the overall scenario regarding the impact of biophysical parameters on LST. Spearman’s correlation coefficient (r) was used to understand the relationship between LST and the biophysical indices. The biophysical indices have a significant impact on the LST [25], and the pattern of the thermal environment in urban areas varies spatially due to variation in the surface properties [62]. Therefore, it is essential to understand the relationship between LST and biophysical parameters. PCA was also used to identify the factors affecting LST. All the statistical analyses were carried out using SPSS software (version 22).

## 3. Results 

### 3.1. LULC Dynamics in Jeddah

In Jeddah, there have been substantial variations in LULC over the last 20 years. For example, in 2000, the built-up area totaled 3085 hectares, which increased to 4763.69 hectares in 2010 and 5557.98 hectares in 2021. Thus, the built-up area increased by about 80% from 2000 to 2020. On the other hand, open land totaled 18,494 hectares in 2000, which decreased to 15,782.32 hectares in 2021, representing a decline from 82.85% to 74.96% (Table 2 and Figure 2). From 2000 to 2021, the vegetation area increased from 26.23 to 34.66 hectares (an increase of about 32%). As per the kappa statistic, it was found that the average accuracy of LULC maps was 82%. The spatial distribution of LST in Jeddah megacity is shown in Figure 3.

### 3.2. Pattern of the Biophysical Indices in Jeddah

Table 3 provides the descriptive statistics of the biophysical indices in Jeddah city for the year of 2021. The maps show significant variation in biophysical parameters across the city. The mean value MNDWI was found to be −0.138, with a maximum, minimum, and standard deviation of 0.309, −0.479, and 0.0485, respectively. The mean NDVI value was 0.041, with a maximum, minimum, and standard deviation of 0.4328, −0.399, and 0.0248, respectively. In the case of NDBI, the mean value was 0.048, with a maximum, minimum, and standard deviation of 0.461, −0.342, and 0.048, respectively. In the case of the spatial variation of the biophysical indices, it was found that the areas with high NDVI are mainly scattered. High values of NDVI were mainly concentrated in the middle–western parts of the city (along the sea). High NDVI values were mainly observed in the northern, southern, and middle eastern parts of the city. These areas are characterized by open impervious surfaces. In the case of NDWI, the areas along the sea are characterized by high NDWI values. The areas in the southern, northern, and middle eastern parts of the city are characterized by having the lowest NDWI and bare impervious surfaces. The details of NDBaI, SAVI, and MNDWI are presented in the maps (Figure 4).

### 3.3. Impact of Biophysical Parameters on the LST

The biophysical indices have a significant impact on the LST [44]. In this study, the correlation between LST and biophysical indices was determined using Spearman’s correlation coefficient (r) (Table 4). The correlation results showed that LST has a positive relationship with NDBI (r = 0.665) and NDBaI (r = 0.367), and a very weak positive relationship with NDVI and SAVI. On the other hand, MNDWI and NDWI have a negative relationship with LST (Figure 5). This result clearly indicates that the areas with higher water coverage are characterized by lower LST. These findings agree well with previous literature.

### 3.4. Factors Affecting LST

In this study, principal component analysis (PCA) was used to understand the factors affecting LST in Jeddah in terms of remote-sensing-based biophysical indices (Table 5). The results show that factor 1 could be explained by two indices: NDVI and SAVI. These indices are related to the vegetation cover, clearing demonstrating that it plays a significant role in determining the LST in the city. In previous studies, it was commonly reported that areas with higher vegetation cover are characterized by lower LST and vice versa [63,64]. Thus, vegetation cover (green space) had significant impact on the cooling effect in urban areas [65]. Factor 1 explained 70% of the total variance out of the five factors considered in this study. In the case of factor 2, there were also two indices: NDBI and NDBaI. Factor 2 explained 20.72% of the variance. These indices were found to be strongly affected by human activities. In the previous studies, it was documented that impervious surface areas are characterized by high LST. These two factors explained more than 90% of the total variance in the study.

## 4. Discussion

Thus, from the results, it can be clearly recognized that built-up areas were characterized by higher LST than the areas covered with vegetation and water bodies. Ren et al. [66] carried out a study in Zhengzhou (China) using the local climatic zones (LCZ) approach, and their results showed that built-up areas were characterized by higher LST than the natural landscapes (such as vegetation cover and water bodies). A similar kind of result was reported by Chen et al. [67] in Liaoning Province (northeast China). The spatial variation of the thermal pattern in urban areas is influenced by a number of factors, such as environmental, social and urban forms [67,68]. Kurniati and Nitivattananon [69], carried out a study in Surabaya city (Indonesia) and stated that the surface properties and urban forms of the urban areas, such as green cover use of asphalt, largely influence thermal behaviour. A study was performed by Shi et al. [68] in the high-density city of Guangzhou (China), and their findings showed that vegetation cover had a negative and building density had a positive correlation on LST. In this study, although building density has not been considered to find out the relationship with LST, NDBI showed that there was a positive correlation between LST and NDBI. Similar findings werealso reported in other previous studies [70,71]

Thus, based on the previous studies, it is well documented that socioeconomic, environmental, and spatial forms largely influence thermal patterns in an urban environment. Therefore, urgent action needs to be taken to ensure the sustainability of urban environmental development [72,73]. A continuous increase in LST ultimately leads to the emergence of UHI effects [74,75] Therefore, city planners and policy makers must adopt technical and nature-based solutions (NBS), such as innovative urban landscape planning and adoption strategies related to green infrastructure, to mitigate the effects of UHIs. Previous studies have examined a number of strategies that were implemented to mitigate UHI effects, such as the development of green infrastructure [76,77] and applications of highly reflective pavements [78] and sustainable urban morphological design [79], and found that they largely mitigate UHI effects by modifying surface energy [80]. The effectiveness of UHI mitigation depends on the eco-environmental conditions of urban areas. For example, areas with green infrastructure act as sink areas for temperature, and impervious surface areas act as sources of UHI phenomena [81].

Among all these strategies, NBS has been recognized as one of the most significant measures to mitigate UHI effects [82,83,84]. NBS was successfully implemented in urban areas to mitigate a series of urban challenges such as climate change and UHI phenomena [85]. In the context of Saudi cities, an emphasis should be placed on urban greening to mitigate UHI effects. The applications of green roofs, vertical greenery, and green facades have emerged as some of the most significant strategies to mitigate UHI effects [86,87]. In many previous studies, the relationship between green infrastructure and local temperature was explored to design mitigation strategies [70]. The impacts of green infrastructure have been well documented in previous literature [88,89]. The areas with higher impervious surfaces were characterized by higher LST, indicating that these areas are subject to strong UHI effects. Therefore, in these areas, greenness can be increased to reduce the impact of UHI and mitigate heat stress in a city.

Although this study showed an interesting assessment to understand the relationship between LST and biophysical indices in a megacity located in a hot desert climate, it has a few limitations, e.g., this study has not assessed the relationship between LST and biophysical indices on a temporal basis. Therefore, in a future study, temporal analysis can be more effective to help us understand the relationship between LST and biophysical indices. Secondly, a few remote sensing indices were used to find out the relationship between LST and biophysical indices. In future studies, more remote sensing indices can be utilized for a better interpretation of the relationship.

## 5. Conclusions

In this study, an attempt was made to understand the relationship between biophysical indices and land surface temperature (LST) in a hot desert megacity (Jeddah) in Saudi Arabia using remote sensing. Six remote-sensing-based indices were used to understand the relationship between biophysical indices and LST in this Saudi megacity, and a few notable findings were recorded. First, from 2000 to 2021, the built-up area in Jeddah megacity increased by about 80% while open or bare area decreased from 82.85% to 74.96%. Second, there were substantial spatial variations in biophysical indices across the megacity, and most parts of the megacity were found to be characterized by impervious surface (covered with built-up and bare or open land). Third, NDBI demonstrated a strong positive correlation with LST, indicating that the areas with high NDBI are characterized by high LST. On the other hand, MNDWI and NDVI were negatively and very poorly positively correlated with LST, respectively. This result clearly indicates that reductions in LST in the megacity could be achieved by expanding green infrastructure. Fourth, the PCA results show that vegetation cover plays a significant role in affecting LST in Jeddah. In addition to this, the imperviousness of the surface cover also influences LST in Jeddah megacity.

Thus, based on the overall results, we determined that most of the areas in Jeddah megacity are characterized by very high LST. Moreover, urban green infrastructure can decrease LST, based on the very poor positive correlation with NDVI. These analyses suggest the protective implications of green infrastructure strategies that are needed to mitigate increased LST in Jeddah megacity. In many developed and developing countries, the implications of urban green infrastructures have been largely promoted to mitigate heat in urban areas. Urban green infrastructure planning strategies have been prioritized in particular, for example in Copenhagen [70], Nanjing [88], and Berlin [89]. Thus, green infrastructures can be a very effective measure to mitigate heat. In Saudi Arabia, the Ministry of Municipal and Rural Affairs (MoMRA) has adopted a number of strategies to improve green spaces in cities. The MoMRA has invested huge financial support in the sustainable planning of green spaces in Saudi Arabia. Therefore, green spaces at the city as well as neighborhood scale need to be planned and designed properly.

## Figures and Tables

**Figure 1 ijerph-20-05025-f001:**
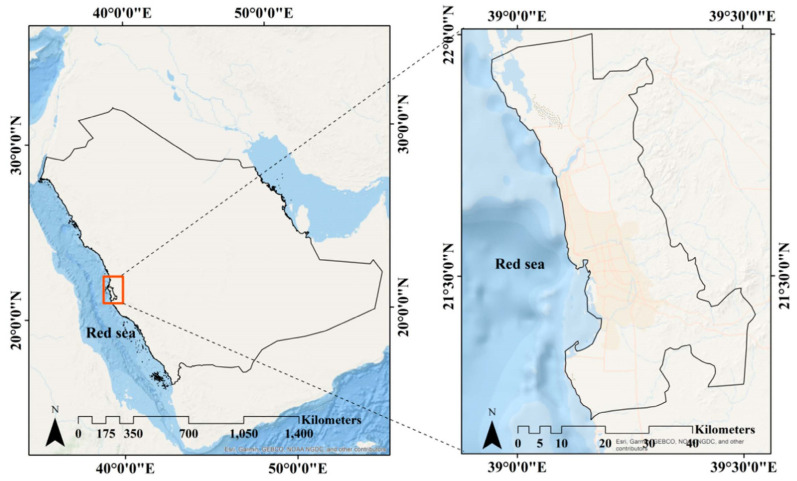
Location map of the study area (Jeddah megacity in Saudi Arabia).

**Figure 2 ijerph-20-05025-f002:**
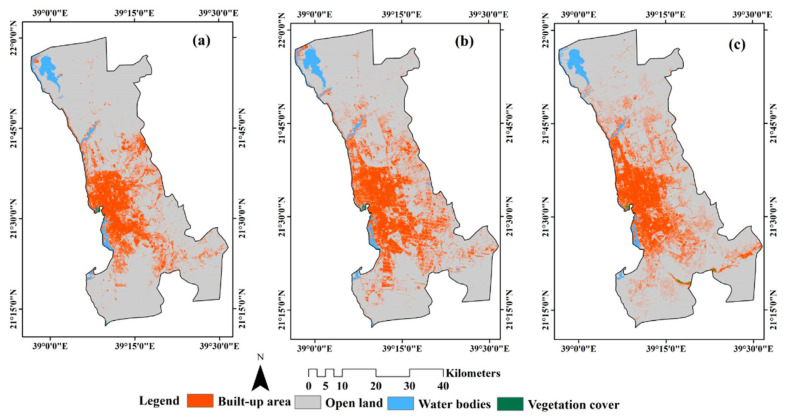
Land use and land cover (LULC) in Jeddah: (**a**) 2000, (**b**) 2010, and (**c**) 2021.

**Figure 3 ijerph-20-05025-f003:**
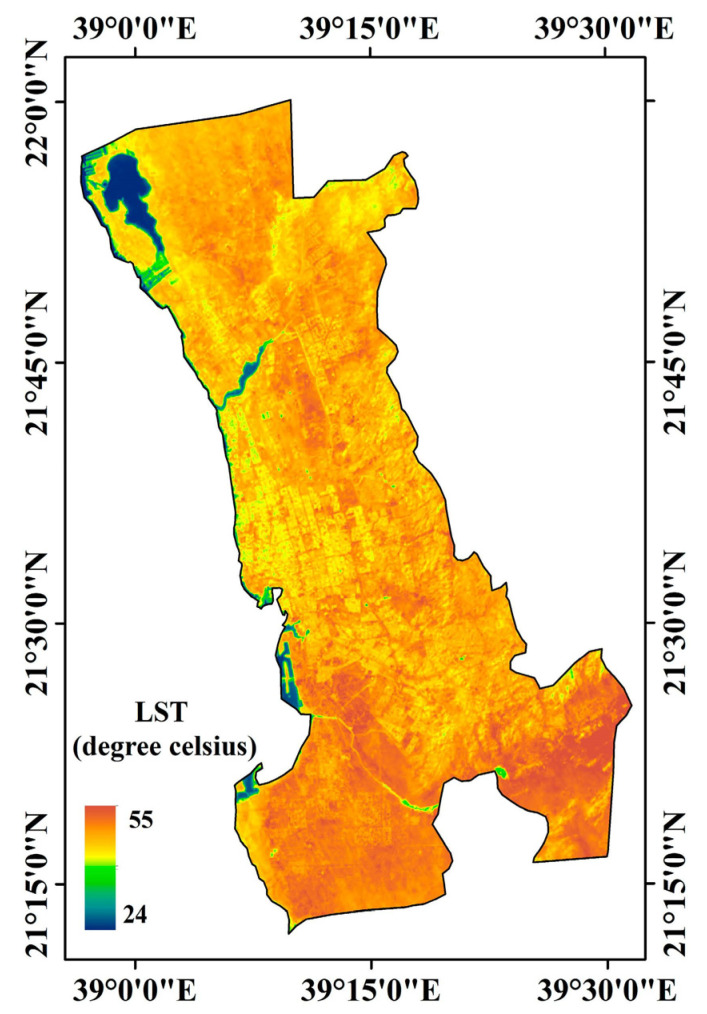
Spatial distribution of LST in Jeddah megacity (2021).

**Figure 4 ijerph-20-05025-f004:**
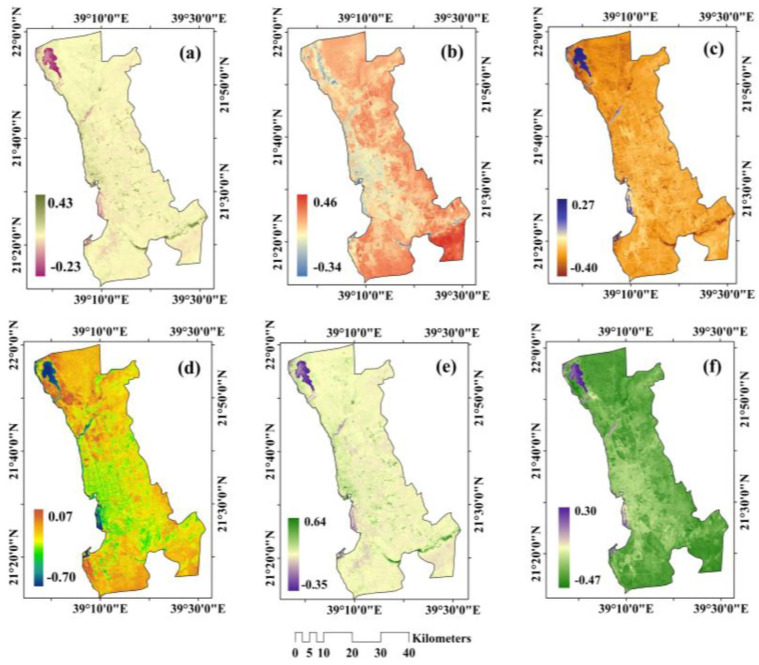
The distribution of biophysical indices in Jeddah (2021): (**a**) NDVI, (**b**) NDBI, (**c**) NDWI, (**d**) NDBaI, (**e**) SAVI, and (**f**) MNDWI (2021).

**Figure 5 ijerph-20-05025-f005:**
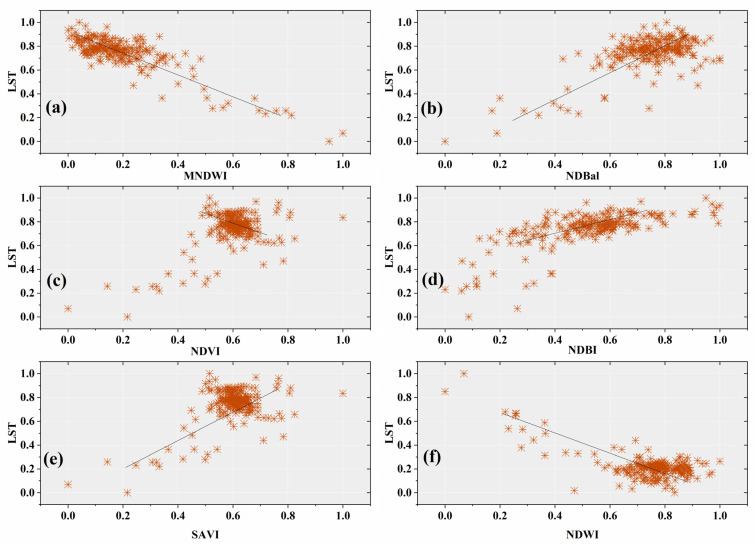
Scatter plots showing the relationship between LST and biophysical indices (2021) (**a**) MNDWI (**b**) NDBaI (**c**) NDVI (**d**) NDBI (**e**) SAVI (**f**) MDWI.

**Table 1 ijerph-20-05025-t001:** Details of the data sources used in this study.

Year	Date of Acquisition	Path/Row	Sensor	Source	Purpose
2000	30 September 2000	170/45	LANDSAT TM	USGS (https://earthexplorer.usgs.gov/)	LULC map
2010	2 April 2010		LANDSAT TM		LULC map
2021	16 September 2021		LANDSAT OLI		LULC map and biophysical indices

**Table 2 ijerph-20-05025-t002:** LULC in Jeddah megacity for the indicated years.

LULC Type	2000	%	2010	%	2021	%
Built up	3085.64	14.05	4763.96	19.85	5557.98	22.12
Open land	18,191.44	82.85	16,519.87	76.91	15,782.32	74.96
Water	653.51	2.98	673.47	3.14	578.46	2.75
Vegetation	26.23	0.12	22.66	0.11	34.66	0.16

**Table 3 ijerph-20-05025-t003:** Descriptive statistics of the biophysical indices.

Biophysical Indices	Max	Min	Mean	SD
MNDWI	0.309	−0.479	−0.138	0.0485
NDBI	0.461	−0.342	0.048	0.0289
NDVI	0.431	−0.238	0.0401	0.0248
NDWI	0.277	−0.399	−0.09	0.0377
SAVI	0.648	−0.057	0.06	0.0372

**Table 4 ijerph-20-05025-t004:** Correlation between LST and the biophysical indices.

Biophysical Indices	MNDWI	NDBaI	NDVI	NDBI	SAVI	NDWI	LST
MNDWI	1.000	−0.796 **	−0.160 *	−0.848 **	−0.157 *	0.580 **	−0.620 **
NDBaI	−0.796 **	1.000	0.138 *	0.588 **	0.131	−0.601 **	0.367 **
NDVI	−0.160 *	0.138 *	1.000	−0.143 *	0.998 **	−0.689 **	0.049
NDBI	−0.848 **	0.588 **	−0.143 *	1.000	−0.146 *	−0.204 **	0.665 **
SAVI	−0.157 *	0.131	0.998 **	−0.146 *	1.000	−0.688 **	0.052
NDWI	0.580 **	−0.601 **	−0.689 **	−0.204 **	−0.688 **	1.000	−0.203 **
LST	−0.620 **	0.367 **	0.049	0.665 **	0.052	−0.203 **	1.000

** Correlation is significant at a 0.01 confidence level (2-tailed). * Correlation is significant at a 0.05 confidence level (2-tailed).

**Table 5 ijerph-20-05025-t005:** PCA analysis of factors.

Biophysical Indices	Component				
	1	2	3	4	5
MNDWI	−0.448	−0.684	−0.504	−0.226	0.160
NDBaI	0.295	0.423	0.846	0.134	0.018
NDVI	0.968	0.075	0.198	0.135	−0.007
NDBI	−0.043	0.957	0.244	0.146	0.003
SAVI	0.968	0.076	0.195	0.140	−0.011
NDWI	−0.706	−0.238	−0.564	−0.222	0.278
LST	0.391	0.581	0.247	0.669	−0.032
Variance explained (%)	70.42	20.72	5.62	2.29	0.88
Cumulative (%)	70.42	91.15	96.77	99.07	99.95

## Data Availability

The data that support the findings of this study are available from the corresponding author, Abdullah Addas, upon reasonable request.

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
