# Peer review of "Understanding the Relationship between Urban Biophysical Composition and Land Surface Temperature in a Hot Desert Megacity (Saudi Arabia)"

_ijerph, 2023, doi:10.3390/ijerph20065025_

Round 1

Author Response

Dear Reviewer 

Thank you for your comments and suggestion, please see the attached file.

Regards 

Reviewer 2 Report

First of all, I would like to congratulate the author for the fantastic manuscript produced.

The manuscript “Understanding the Relationship between Urban Biophysical Composition and Land Surface Temperature in a Hot Desert Megacity (Saudi Arabia)”; is of great interest for publication in International Journal of Environmental Research and Public Health. The topic analyzed is currently of great interest and may be of interest to apply it in other urban spaces. The article can be accepted for publication after a few changes.

I recommend differentiating the results and discussion sections in order to respect the classical structure of an investigation of this type.

On figure 2. Include the letter (b) in the middle image. I also recommend the author to try to improve the scale, to better identify "vegetation cover".

On figures 1, 2 and 3, the same font size should be respected.

The results that the work shows are of great interest, in this sense I suggest that in the discussion or in the conclusions, some deeper reflection could be introduced about results, climate change and adaptation strategies to be developed or that are being considered to be developed in the study area. In this sense are important incorporate elements such as the sixth report of the IPCC.

In general, all the sections are clearly developed and well written, making use of current bibliography related to the subject matter.

Finally, congratulations to the author for the great work done.

Author Response

(The authors gave the same response as above.)

Round 2

Reviewer 1 Report

I would like to thank the author for taking the time in addressing the reviewer's comments. The manuscript has improved. It is acceptable for publication after addressing the following comments. 

1) The author did not clarify my comment#10 from the earlier review. If Figure 5 and Table 3 show the statistics between the LST and the biophysical indices, how come there are no negative values (the x-axis of all the subplots in Figure 5 ranges from 0 to 1) for the indicators? For example, if you consider the indicator MNDWI, it is mentioned that the mean value is -0.138, and it has maximum, and minimum values of 0.309, and -0.479 respectively. Now, if one looks at the corresponding scatter plot from Figure 5, that is subplot 5(a) drawn between LST vs MNDWI, there are no negative MNDWI values seen in the plot. If all of the indicators used in the study have a range from -1 to +1 as mentioned in the writing, how come Figure 5 has all positive indicators but Table 3 has negative minimums? Also, why is the y-axis LST has a range of 0 to 1? If the x-axis variables and the LST values were normalized by their absolute maximum values, it is not mentioned anywhere in the text. More discussion is required on how Figure 5 was drawn, or plot it using the actual values. For reference, if the figure represents the actual LST vs indicators value in the form of a scatter plot, I expected it to look similar to Figure 8 in Abulibdeh(2021), where the LST (y-axis) has a reasonable range (20 to 40 degree Celsius) and the indicators (for example, NDBI) varied from -0.3 to 0.3 for Jeddah region.

2) Instead of giving the legend values as "High" and "Low" for Figures 3 and 4, please use actual values so that the reader can understand the spatial distribution. The actual temperature values for the study region are not given anywhere in the text. Figure 3 would be a good place to show this.

References used:

Abulibdeh, A., 2021. Analysis of urban heat island characteristics and mitigation strategies for eight arid and semi-arid gulf region cities. Environmental Earth Sciences, 80, pp.1-26. 

Author Response

Dear Reviewer 

Thank you very much for your valuable comments, please see the attached file.

Regards 
